# An Inner-loop Free Solution to Inverse Problems using Deep Neural Networks

**Kai Fai**[*]
Duke University
`kai.fan@stat.duke.edu`

**Qi Wei**[*]
Duke University
`qi.wei@duke.edu`

**Lawrence Carin**
Duke University
`lcarin@duke.edu`

**Katherine Heller**
Duke University
`kheller@stat.duke.edu`

## Abstract

We propose a new method that uses deep learning techniques to accelerate the popular alternating direction method of multipliers (ADMM) solution for inverse problems. The ADMM updates consist of a proximity operator, a least squares regression that includes a big matrix inversion, and an explicit solution for updating the dual variables. Typically, inner loops are required to solve the first two sub-minimization problems due to the intractability of the prior and the matrix inversion. To avoid such drawbacks or limitations, we propose an *inner-loop free* update rule with two pre-trained deep convolutional architectures. More specifically, we learn a conditional denoising auto-encoder which imposes an implicit data-dependent prior/regularization on ground-truth in the first sub-minimization problem. This design follows an empirical Bayesian strategy, leading to so-called amortized inference. For matrix inversion in the second sub-problem, we learn a convolutional neural network to approximate the matrix inversion, i.e., the inverse mapping is learned by feeding the input through the learned forward network. Note that training this neural network does not require ground-truth or measurements, i.e., data-independent. Extensive experiments on both synthetic data and real datasets demonstrate the efficiency and accuracy of the proposed method compared with the conventional ADMM solution using inner loops for solving inverse problems.

## 1 Introduction

Most of the inverse problems are formulated directly to the setting of an optimization problem related to the a forward model [25]. The forward model maps unknown signals, i.e., the ground-truth, to acquired information about them, which we call data or measurements. This mapping, or forward problem, generally depends on a physical theory that links the ground-truth to the measurements. Solving inverse problems involves learning the inverse mapping from the measurements to the ground-truth. Specifically, it recovers a signal from a small number of degraded or noisy measurements. This is usually ill-posed [26, 25]. Recently, deep learning techniques have emerged as excellent models and gained great popularity for their widespread success in allowing for efficient inference techniques on applications include pattern analysis (unsupervised), classification (supervised), computer vision, image processing, etc [6]. Exploiting deep neural networks to help solve inverse problems has been explored recently [24, 1] and deep learning based methods have achieved state-of-the-art performance in many challenging inverse problems like super-resolution [3, 24], image reconstruction [20],

---

[*]The authors contributed equally to this work.

automatic colorization [13]. More specifically, massive datasets currently enables learning end-to-end mappings from the measurement domain to the target image/signal/data domain to help deal with these challenging problems instead of solving the inverse problem by inference. This mapping function from degraded data point to ground-truth has recently been characterized by using sophisticated networks, e.g., deep neural networks. A strong motivation to use neural networks stems from the universal approximation theorem [5], which states that a feed-forward network with a single hidden layer containing a finite number of neurons can approximate any continuous function on compact subsets of $\mathbb{R}^n$, under mild assumptions on the activation function.

More specifically, in recent work [3, 24, 13, 20], an end-to-end mapping from measurements $\mathbf{y}$ to ground-truth $\mathbf{x}$ was learned from the training data and then applied to the testing data. Thus, the complicated inference scheme needed in the conventional inverse problem solver was replaced by feeding a new measurement through the pre-trained network, which is much more efficient. To improve the scope of deep neural network models, more recently, in [4], a splitting strategy was proposed to decompose an inverse problem into two optimization problems, where one sub-problem, related to regularization, can be solved efficiently using trained deep neural networks, leading to an alternating direction method of multipliers (ADMM) framework [2, 17]. This method involves training a deep convolutional auto-encoder network for low-level image modeling, which explicitly imposes regularization that spans the subspace that the ground-truth images live in. For the sub-problem that requires inverting a big matrix, a conventional gradient descent algorithm was used, leading to an alternating update, iterating between feed-forward propagation through a network and iterative gradient descent. Thus, an inner loop for gradient descent is still necessary in this framework. A similar approach to learn approximate ISTA with neural network is illustrated in [11].

In this work, we propose an inner-loop free framework, in the sense that no iterative algorithm is required to solve sub-problems, using a splitting strategy for inverse problems. The alternating updates for the two sub-problems were derived by feeding through two pre-trained deep neural networks, i.e., one using an amortized inference based denoising convolutional auto-encoder network for the proximity operation and one using structured convolutional neural networks for the huge matrix inversion related to the forward model. Thus, the computational complexity of each iteration in ADMM is linear with respect to the dimensionality of the signals. The network for the proximity operation imposes an implicit prior learned from the training data, including the measurements as well as the ground-truth, leading to amortized inference. The network for matrix inversion is independent from the training data and can be trained from noise, i.e., a random noise image and its output from the forward model. To make training the networks for the proximity operation easier, three tricks have been employed: the first one is to use a pixel shuffling technique to equalize the dimensionality of the measurements and ground-truth; the second one is to optionally add an adversarial loss borrowed from the GAN (Generative Adversarial Nets) framework [10] for sharp image generation; the last one is to introduce a perceptual measurement loss derived from pre-trained networks, such as AlexNet [12] or VGG-16 Model [23]. Arguably, the speed of the proposed algorithm, which we term Inf-ADMM-ADNN (*Inner-loop free ADMM with Auxiliary Deep Neural Network*), comes from the fact that it uses two auxiliary pre-trained networks to accelerate the updates of ADMM.

**Contribution** The main contribution of this paper is comprised of **i**) learning an implicit prior/regularizer using a denoising auto-encoder neural network, based on amortized inference; **ii**) learning the inverse of a big matrix using structured convolutional neural networks, without using training data; **iii**) each of the above networks can be exploited to accelerate the existing ADMM solver for inverse problems.

## 2   Linear Inverse Problem

**Notation**: trainable networks by calligraphic font, e.g., $\mathcal{A}$, fixed networks by italic font e.g., $A$. As mentioned in the last section, the low dimensional measurement is denoted as $\mathbf{y} \in \mathbb{R}^m$, which is reduced from high dimensional ground truth $\mathbf{x} \in \mathbb{R}^n$ by a linear operator $A$ such that $\mathbf{y} = A\mathbf{x}$. Note that usually $n \geq m$, which makes the number of parameters to estimate no smaller than the number of data points in hand. This imposes an ill-posed problem for finding solution $\mathbf{x}$ on new observation $\mathbf{y}$, since $A$ is an underdetermined measurement matrix. For example, in a super-resolution set-up, the matrix $A$ might not be invertible, such as the strided Gaussian convolution in [21, 24]. To overcome this difficulty, several computational strategies, including Markov chain Monte Carlo (MCMC) and tailored variable splitting under the ADMM framework, have been proposed and applied to different

kinds of priors, e.g., the empirical Gaussian prior [29, 32], the Total Variation prior [22, 30, 31], etc. In this paper, we focus on the popular ADMM framework due to its low computational complexity and recent success in solving large scale optimization problems. More specifically, the optimization problem is formulated as

$$\hat{\mathbf{x}} = \arg\min_{\mathbf{x}, \mathbf{z}} \|\mathbf{y} - A\mathbf{z}\|^2 + \lambda \mathcal{R}(\mathbf{x}), \quad s.t. \quad \mathbf{z} = \mathbf{x} \tag{1}$$

where the introduced auxiliary variable $\mathbf{z}$ is constrained to be equal to $\mathbf{x}$, and $\mathcal{R}(\mathbf{x})$ captures the structure promoted by the prior/regularization. If we design the regularization in an empirical Bayesian way, by imposing an implicit data dependent prior on $\mathbf{x}$, i.e., $\mathcal{R}(\mathbf{x}; \mathbf{y})$ for amortized inference [24], the augmented Lagrangian for (1) is

$$\mathcal{L}(\mathbf{x}, \mathbf{z}, \mathbf{u}) = \|\mathbf{y} - A\mathbf{z}\|^2 + \lambda \mathcal{R}(\mathbf{x}; \mathbf{y}) + \langle \mathbf{u}, \mathbf{x} - \mathbf{z} \rangle + \beta \|\mathbf{x} - \mathbf{z}\|^2 \tag{2}$$

where $\mathbf{u}$ is the Lagrange multiplier, and $\beta > 0$ is the penalty parameter. The usual augmented Lagrange multiplier method is to minimize $\mathcal{L}$ w.r.t. $\mathbf{x}$ and $\mathbf{z}$ simultaneously. This is difficult and does not exploit the fact that the objective function is separable. To remedy this issue, ADMM decomposes the minimization into two subproblems that are minimizations w.r.t. $\mathbf{x}$ and $\mathbf{z}$, respectively. More specifically, the iterations are as follows:

$$\mathbf{x}^{k+1} = \arg\min_{\mathbf{x}} \beta \|\mathbf{x} - \mathbf{z}^k + \mathbf{u}^k/2\beta\|^2 + \lambda \mathcal{R}(\mathbf{x}; \mathbf{y}) \tag{3}$$

$$\mathbf{z}^{k+1} = \arg\min_{\mathbf{z}} \|\mathbf{y} - A\mathbf{z}\|^2 + \beta \|\mathbf{x}^{k+1} - \mathbf{z} + \mathbf{u}^k/2\beta\|^2 \tag{4}$$

$$\mathbf{u}^{k+1} = \mathbf{u}^k + 2\beta(\mathbf{x}^{k+1} - \mathbf{z}^{k+1}). \tag{5}$$

If the prior $\mathcal{R}$ is appropriately chosen, such as $\|\mathbf{x}\|_1$, a closed-form solution for (3), i.e., a soft thresholding solution is naturally desirable. However, for some more complicated regularizations, e.g., a patch based prior [8], solving (3) is nontrivial, and may require iterative methods. To solve (4), a matrix inversion is necessary, for which conjugate gradient descent (CG) is usually applied to update $\mathbf{z}$ [4]. Thus, solving (3) and (4) is in general cumbersome. Inner loops are required to solve these two sub-minimization problems due to the intractability of the prior and the inversion, resulting in large computational complexity. To avoid such drawbacks or limitations, we propose an *inner loop-free* update rule with two pretrained deep convolutional architectures.

## 3 Inner-loop free ADMM

### 3.1 Amortized inference for $\mathbf{x}$ using a conditional proximity operator

Solving sub-problem (3) is equivalent to finding the solution of the **proximity operator** $\mathcal{P}_{\mathcal{R}}(\mathbf{v}; \mathbf{y}) = \arg\min_{\mathbf{x}} \frac{1}{2}\|\mathbf{x} - \mathbf{v}\|^2 + \mathcal{R}(\mathbf{x}; \mathbf{y})$, where we incorporate the constant $\frac{\lambda}{2\beta}$ into $\mathcal{R}$ without loss of generality. If we impose the first order necessary conditions [18], we have

$$\mathbf{x} = \mathcal{P}_{\mathcal{R}}(\mathbf{v}; \mathbf{y}) \Leftrightarrow 0 \in \partial \mathcal{R}(\cdot; \mathbf{y})(\mathbf{x}) + \mathbf{x} - \mathbf{v} \Leftrightarrow \mathbf{v} - \mathbf{x} \in \partial \mathcal{R}(\cdot; \mathbf{y})(\mathbf{x}) \tag{6}$$

where $\partial \mathcal{R}(\cdot; \mathbf{y})$ is a partial derivative operator. For notational simplicity, we define another operator $\mathcal{F} =: \mathcal{I} + \partial \mathcal{R}(\cdot; \mathbf{y})$. Thus, the last condition in (6) indicates that $\mathbf{x}^{k+1} = \mathcal{F}^{-1}(\mathbf{v})$. Note that the inverse here represents the inverse of an operator, i.e., the inverse function of $\mathcal{F}$. Thus our objective is to learn such an inverse operator which projects $\mathbf{v}$ into the prior subspace. For simple priors like $\|\cdot\|_1$ or $\|\cdot\|_2^2$, the projection can be efficiently computed. In this work, we propose an implicit example-based prior, which does not have a truly Bayesian interpretation, but aids in model optimization. In line with this prior, we define the implicit proximity operator $\mathcal{G}_{\boldsymbol{\theta}}(\mathbf{x}; \mathbf{v}, \mathbf{y})$ parameterized by $\boldsymbol{\theta}$ to approximate unknown $\mathcal{F}^{-1}$. More specifically, we propose a neural network architecture referred to as conditional Pixel Shuffling Denoising Auto-Encoders (cPSDAE) as the operator $\mathcal{G}$, where pixel shuffling [21] means periodically reordering the pixels in each channel mapping a high resolution image to a low resolution image with scale $r$ and increase the number of channels to $r^2$ (see [21] for more details). This allows us to transform $\mathbf{v}$ so that it is the same scale as $\mathbf{y}$, and concatenate it with $\mathbf{y}$ as the input of cPSDAE easily. The architecture of cPSDAE is shown in Fig. 1 (d).

### 3.2 Inversion-free update of $\mathbf{z}$

While it is straightforward to write down the closed-form solution for sub-problem (4) w.r.t. $\mathbf{z}$ as is shown in (7), explicitly computing this solution is nontrivial.

$$\mathbf{z}^{k+1} = K\left(A^\top \mathbf{y} + \beta \mathbf{x}^{k+1} + \mathbf{u}^k/2\right), \text{ where } K = \left(A^\top A + \beta \mathbf{I}\right)^{-1} \tag{7}$$

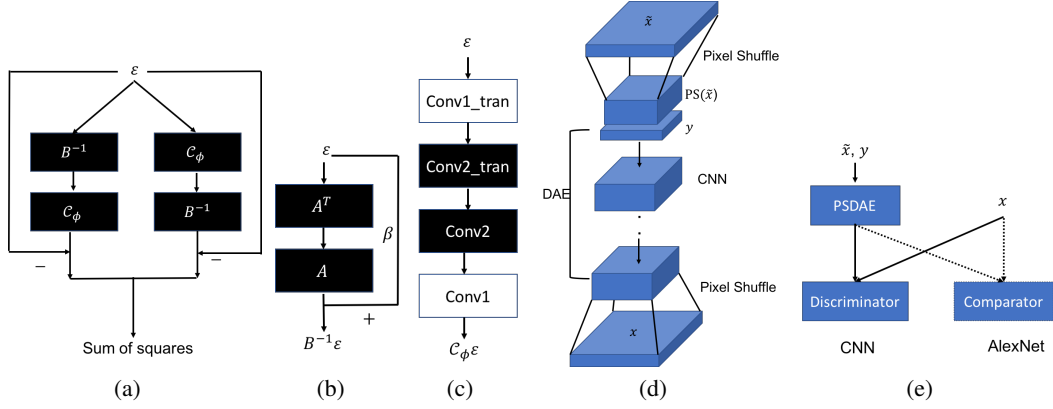

Figure 1: Network for updating $\mathbf{z}$ (in black): (a) loss function (9), (b) structure of $B^{-1}$, (c) struture of $\mathcal{C}_{\phi}$. Note that the input $\epsilon$ is random noise independent from the training data. Network for updating $\mathbf{z}$ (in blue): (d) structure of cPSDAE $\mathcal{G}_{\theta}(\mathbf{x}; \tilde{\mathbf{x}}, \mathbf{y})$ ($\tilde{\mathbf{x}}$ plays the same role as $\mathbf{v}$ in training), (e) adversarial training for $\mathcal{R}(\mathbf{x}; \mathbf{y})$. Note again that (a)(b)(c) describes the network for inferring $\mathbf{z}$, which is data-independent and (d)(e) describes the network for inferring $\mathbf{x}$, which is data-dependent.

In (7), $A^{\top}$ is the transpose of the matrix $A$. As we mentioned, the term $K$ in the right hand side involves an expensive matrix inversion with computational complexity $O(n^3)$ . Under some specific assumptions, e.g., $A$ is a circulant matrix, this matrix inversion can be accelerated with a Fast Fourier transformation, which has a complexity of order $\mathcal{O}(n \log n)$. Usually, the gradient based update has linear complexity in each iteration and thus has an overall complexity of order $\mathcal{O}(n_{\text{int}} \log n)$, where $n_{\text{int}}$ is the number of iterations. In this work, we will learn this matrix inversion explicitly by designing a neural network. Note that $K$ is only dependent on $A$, and thus can be computed in advance for future use. This problem can be reduced to a smaller scale matrix inversion by applying the Sherman-Morrison-Woodbury formula:

$$K = \beta^{-1} \left( \mathbf{I} - A^{\top} B A \right), \text{ where } B = \left( \beta \mathbf{I} + AA^{\top} \right)^{-1}. \tag{8}$$

Therefore, we only need to solve the matrix inversion in dimension $m \times m$, i.e., estimating $B$. We propose an approach to approximate it by a trainable deep convolutional neural network $\mathcal{C}_{\phi} \approx B$ parameterized by $\phi$. Note that $B^{-1} = \lambda \mathbf{I} + AA^{\top}$ can be considered as a two-layer fully-connected or convolutional network as well, but with a fixed kernel. This inspires us to design two auto-encoders with shared weights, and minimize the sum of two reconstruction losses to learn the inversion $\mathcal{C}_{\phi}$ :

$$\arg \min_{\phi} \mathbb{E}_{\boldsymbol{\varepsilon}} \left[ \| \boldsymbol{\varepsilon} - \mathcal{C}_{\phi} B^{-1} \boldsymbol{\varepsilon} \|_2^2 + \| \boldsymbol{\varepsilon} - B^{-1} \mathcal{C}_{\phi} \boldsymbol{\varepsilon} \|_2^2 \right] \tag{9}$$

where $\boldsymbol{\varepsilon}$ is sampled from a standard Gaussian distribution. The loss in (9) is clearly depicted in Fig. 1 (a) with the structure of $B^{-1}$ in Fig. 1 (b) and the structure of $\mathcal{C}_{\phi}$ in Fig. 1 (c). Since the matrix $B$ is symmetric, we can reparameterize $\mathcal{C}_{\phi}$ as $\mathcal{W}_{\phi} \mathcal{W}_{\phi}^{\top}$, where $\mathcal{W}_{\phi}$ represents a multi-layer convolutional network and $\mathcal{W}_{\phi}^{\top}$ is a symmetric convolution transpose architecture using shared kernels with $\mathcal{W}_{\phi}$, as shown in Fig. 1 (c) (the blocks with the same colors share the same network parameters). By plugging the learned $\mathcal{C}_{\phi}$ in (8) , we obtain a reusable deep neural network $\mathcal{K}_{\phi} = \beta^{-1} \left( \mathbf{I} - A^{\top} \mathcal{C}_{\phi} A \right)$ as a surrogate for the exact inverse matrix $K$. The update of $\mathbf{z}$ at each iteration can be done by applying the same $\mathcal{K}_{\phi}$ as follows:

$$\mathbf{z}^{k+1} \leftarrow \beta^{-1} \left( \mathbf{I} - A^{\top} \mathcal{C}_{\phi} A \right) \left( A^{\top} \mathbf{y} + \beta \mathbf{x}^{k+1} + \mathbf{u}^k / 2 \right). \tag{10}$$

### 3.3 Adversarial training of cPSDAE

In this section, we will describe the proposed adversarial training scheme for cPSDAE to update $\mathbf{x}$. Suppose that we have the paired training dataset $(\mathbf{x}_i, \mathbf{y}_i)_{i=1}^{N}$, a single cPSDAE with the input pair $(\tilde{\mathbf{x}}, \mathbf{y})$ is trying to minimize the reconstruction error $\mathcal{L}_r(\mathcal{G}_{\theta}(\tilde{\mathbf{x}}, \mathbf{y}), \mathbf{x})$, where $\tilde{\mathbf{x}}$ is a corrupted version of $\mathbf{x}$, i.e., $\tilde{\mathbf{x}} = \mathbf{x} + \mathbf{n}$ where $\mathbf{n}$ is random noise. Notice $\mathcal{L}_r$ in traditional DAE is commonly

defined as $\ell_2$ loss, however, $\ell_1$ loss is an alternative in practice. Additionally, we follow the idea in [19, 7] by introducing a discriminator and a comparator to help train the cPSDAE, and find that it can produce sharper or higher quality images than merely optimizing $\mathcal{G}$. This will wrap our conditional generative model $\mathcal{G}_{\boldsymbol{\theta}}$ into the conditional GAN [10] framework with an extra feature matching network (comparator). Recent advances in representation learning problems have shown that the features extracted from well pre-trained neural networks on supervised classification problems can be successfully transferred to others tasks, such as zero-shot learning [15], style transfer learning [9]. Thus, we can simply use pre-trained AlexNet [12] or VGG-16 Model [23] on ImageNet as the comparator without fine-tuning in order to extract features that capture complex and perceptually important properties. The feature matching loss $\mathcal{L}_f(C(\mathcal{G}_{\boldsymbol{\theta}}(\tilde{\mathbf{x}},\mathbf{y})), C(\mathbf{x}))$ is usually the $\ell_2$ distance of high level image features, where $C$ represents the pre-trained network. Since $C$ is fixed, the gradient of this loss can be back-propagated to $\boldsymbol{\theta}$.

For the adversarial training, the discriminator $\mathcal{D}_{\psi}$ is a trainable convolutional network. We can keep the standard discriminator loss as in a traditional GAN, and add the generator loss of the GAN to the previously defined DAE loss and comparator loss. Thus, we can write down our two objectives,

$$\mathcal{L}_D(\mathbf{x},\mathbf{y}) = -\log \mathcal{D}_{\psi}(\mathbf{x}) - \log\left(1 - \mathcal{D}_{\psi}(\mathcal{G}_{\boldsymbol{\theta}}(\tilde{\mathbf{x}},\mathbf{y}))\right) \tag{11}$$

$$\mathcal{L}_G(\mathbf{x},\mathbf{y}) = \lambda_r\|\mathcal{G}_{\boldsymbol{\theta}}(\tilde{\mathbf{x}},\mathbf{y}) - \mathbf{x}\|_2^2 + \lambda_f\|C(\mathcal{G}_{\boldsymbol{\theta}}(\tilde{\mathbf{x}},\mathbf{y})) - C(\mathbf{x})\|_2^2 - \lambda_a \log \mathcal{D}_{\psi}(\mathcal{G}_{\boldsymbol{\theta}}(\tilde{\mathbf{x}},\mathbf{y})) \tag{12}$$

The optimization involves iteratively updating $\psi$ by minimizing $\mathcal{L}_D$ keeping $\boldsymbol{\theta}$ fixed, and then updating $\boldsymbol{\theta}$ by minimizing $\mathcal{L}_G$ keeping $\psi$ fixed. The proposed method, including training and inference has been summarized in Algorithm 1. Note that each update of $\mathbf{x}$ or $\mathbf{z}$ using neural networks in an ADMM iteration has a complexity of linear order w.r.t. the data dimensionality $n$.

### 3.4 Discussion

---
**Algorithm 1** Inner-loop free ADMM with Auxiliary Deep Neural Nets (Inf-ADMM-ADNN)

---
*Training stage*:
1: Train net $\mathcal{K}_{\phi}$ for inverting $A^T A + \beta\mathbf{I}$
2: Train net cPSDAE for proximity operator of $\mathcal{R}(\mathbf{x};\mathbf{y})$

*Testing stage*:
1: **for** $t = 1, 2, \ldots$ **do**
2:     Update $\mathbf{x}$ cf. $\mathbf{x}^{k+1} = \mathcal{F}^{-1}(\mathbf{v})$;
3:     Update $\mathbf{z}$ cf. (10);
4:     Update $\mathbf{u}$ cf. (5);
5: **end for**

---

A critical point for learning-based methods is whether the method generalizes to other problems. More specifically, how does a method that is trained on a specific dataset perform when applied to another dataset? To what extent can we reuse the trained network without re-training?

In the proposed method, two deep neural networks are trained to infer $\mathbf{x}$ and $\mathbf{z}$. For the network w.r.t. $\mathbf{z}$, the training only requires the forward model $A$ to generate the training pairs $(\epsilon, A\epsilon)$. The trained network for $\mathbf{z}$ can be applied for any other datasets as long as $A$ remains the same. Thus, this network can be adapted easily to accelerate inference for inverse problems without training data. However, for inverse problems that depends on a different $A$, a re-trained network is required. It is worth mentioning that the forward model $A$ can be easily learned using training dataset $(\mathbf{x}, \mathbf{y})$, leading to a fully blind estimator associated with the inverse problem. An example of learning $\hat{A}$ can be found in the supplementary materials. For the network w.r.t. $\mathbf{x}$, training requires data pairs $(\mathbf{x}_i, \mathbf{y}_i)$ because of the amortized inference. Note that this is different from training a prior for $\mathbf{x}$ only using training data $\mathbf{x}_i$. Thus, the trained network for $\mathbf{x}$ is confined to the specific tasks constrained by the pairs $(\mathbf{x}, \mathbf{y})$. To extend the generality of the trained network, the amortized setting can be removed, i.e, $\mathbf{y}$ is removed from the training, leading to a solution to proximity operator $\mathcal{P}_{\mathcal{R}}(\mathbf{v}) = \arg\min_{\mathbf{x}} \frac{1}{2}\|\mathbf{x} - \mathbf{v}\|^2 + \mathcal{R}(\mathbf{x})$. This proximity operation can be regarded as a denoiser which projects the noisy version $\mathbf{v}$ of $\mathbf{x}$ into the subspace imposed by $\mathcal{R}(\mathbf{x})$. The trained network (for the proximity operator) can be used as a plug-and-play prior [27] to regularize other inverse problems for datasets that share similar statistical characteristics. However, a significant change in the training dataset, e.g., different modalities like MRI and natural images (e.g., ImageNet [12]), would require re-training.

Another interesting point to mention is the scalability of the proposed method to data of different dimensions. The scalability can be adapted using patch-based methods without loss of generality. For example, a neural network is trained for images of size $64 \times 64$ but the test image is of size $256 \times 256$. To use this pre-trained network, the full image can be decomposed as four $64 \times 64$ images and fed to

the network. To overcome the possible blocking artifacts, eight overlapping patches can be drawn from the full image and fed to the network. The output of these eight patches are then averaged (unweighted or weighted) over the overlapping parts. A similar strategy using patch stitching can be exploited to feed small patches to the network for higher dimensional datasets.

# 4 Experiments

In this section, we provide experimental results and analysis on the proposed Inf-ADMM-ADNN and compare the results with a conventional ADMM using inner loops for inverse problems. Experiments on synthetic data have been implemented to show the fast convergence of our method, which comes from the efficient feed-forward propagation through pre-trained neural networks. Real applications using proposed Inf-ADMM-ADNN have been explored, including single image super-resolution, motion deblurring and joint super-resolution and colorization.

## 4.1 Synthetic data

To evaluate the performance of proposed Inf-ADMM-ADNN, we first test the neural network $\mathcal{K}_\phi$, approximating the matrix inversion on synthetic data. More specifically, we assume that the ground-truth $\mathbf{x}$ is drawn from a Laplace distribution Laplace$(\mu, b)$, where $\mu = 0$ is the location parameter and $b$ is the scale parameter. The forward model $A$ is a sparse matrix representing convolution with a stride of $4$. The architecture of $A$ is available in the supplementary materials (see Section 2). The noise $\mathbf{n}$ is drawn from a standard Gaussian distribution $\mathcal{N}(0, \sigma^2)$. Thus, the observed data is generated as $\mathbf{y} = A\mathbf{x} + \mathbf{n}$. Following Bayes theorem, the maximum a posterior estimate of $\mathbf{x}$ given $\mathbf{y}$, i.e., maximizing $p(\mathbf{x}|\mathbf{y}) \propto p(\mathbf{y}|\mathbf{x})p(\mathbf{x})$ can be equivalently formulated as $\arg\min_{\mathbf{x}} \frac{1}{2\sigma^2}\|\mathbf{y} - A\mathbf{x}\|_2^2 + \frac{1}{b}\|\mathbf{x}\|_1$, where $b = 1$ and $\sigma = 1$ in this setting. Following (3), (4), (5), this problem is reduced to the following three sub-problems: i) $\mathbf{x}^{k+1} = \mathcal{S}_{\frac{1}{2\beta}}(\mathbf{z}^k - \mathbf{u}^k/2\beta)$; ii) $\mathbf{z}^{k+1} = \arg\min_{\mathbf{z}} \|\mathbf{y} - A\mathbf{z}\|_2^2 + \beta\|\mathbf{x}^{k+1} - \mathbf{z} + \mathbf{u}^k/2\beta\|_2^2$; iii) $\mathbf{u}^{k+1} = \mathbf{u}^k + 2\beta(\mathbf{x}^{k+1} - \mathbf{z}^{k+1})$, where the soft thresholding operator $\mathcal{S}$ is defined as $\mathcal{S}_\kappa(a) = \begin{cases} 0 & |a| \leq \kappa \\ a - \mathrm{sgn}(a)\kappa & |a| > \kappa \end{cases}$ and sgn$(a)$ extracts the sign of $a$. The update of $\mathbf{x}^{k+1}$ has a closed-form solution, i.e., soft thresholding of $\mathbf{z}^k - \mathbf{u}^k/2\beta$. The update of $\mathbf{z}^{k+1}$ requires the inversion of a big matrix, which is usually solved using a gradient descent based algorithm. The update of $\mathbf{u}^{k+1}$ is straightforward. Thus, we compare the gradient descent based update, a closed-form solution for matrix inversion[2] and the proposed inner-free update using a pre-trained neural network. The evolution of the objective function w.r.t. the number of iterations and the time has been plotted in the left and middle of Figs. 2. While all three methods perform similarly from iteration to iteration (in the left of Figs. 2), the proposed inner-loop free based and closed-form inversion based methods converge much faster than the gradient based method (in the middle of Figs. 2). Considering the fact that the closed-form solution, i.e., a direct matrix inversion, is usually not available in practice, the learned neural network allows us to approximate the matrix inversion in a very accurate and efficient way.

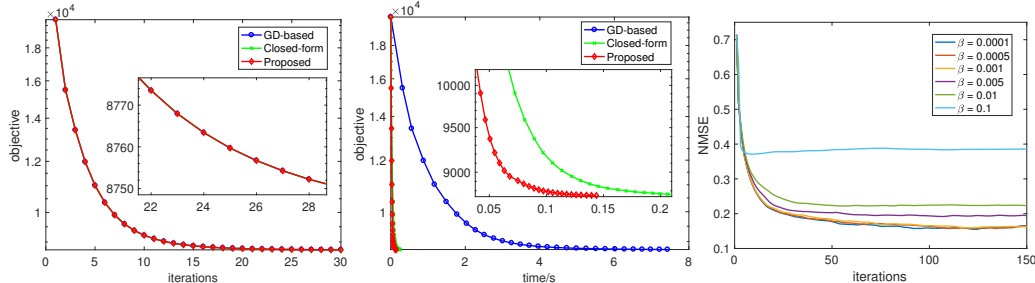

Figure 2: Synthetic data: (left) objective *v.s.* iterations, (middle) objective *v.s.* time. MNIST dataset: (right) NMSE *v.s.* iterations for MNIST image $4\times$ super-resolution.

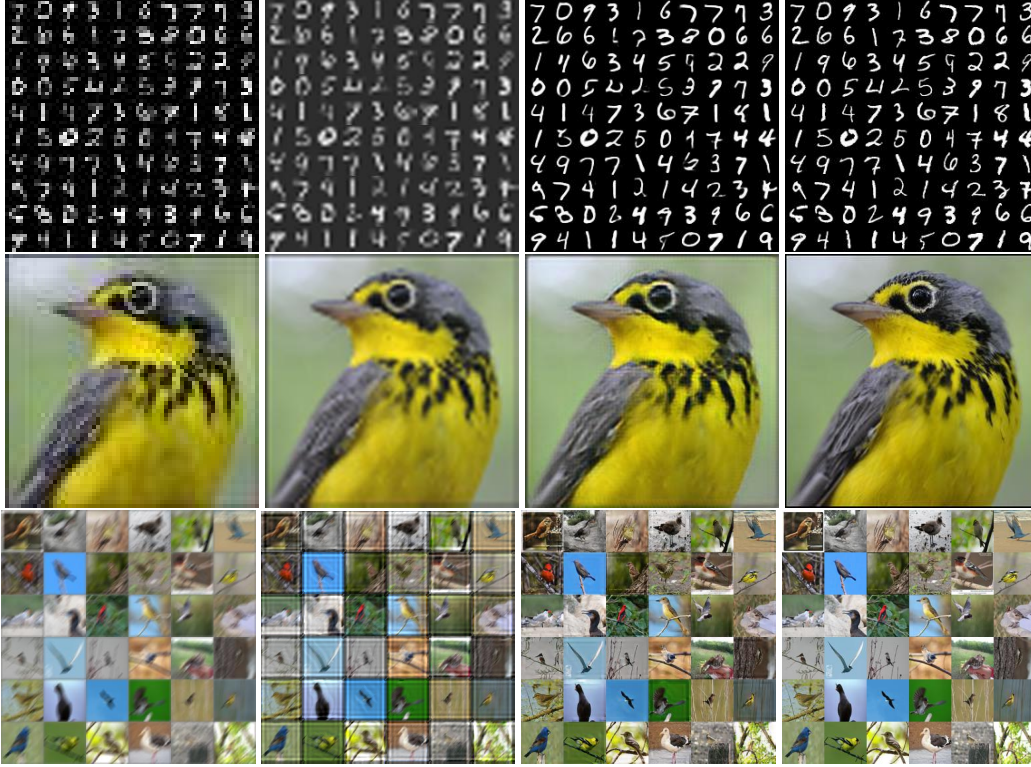

Figure 3: Top two rows : (column 1) LR images, (column 2) bicubic interpolation ($\times 4$), (column 3) results using proposed method ($\times 4$), (column 4) HR image. Bottom row: (column 1) motion blurred images, (column 2) results using Wiener filter with the best performance by tuning regularization parameter, (column 3) results using proposed method, (column 4) ground-truth.

## 4.2 Image super-resolution and motion deblurring

In this section, we apply the proposed Inf-ADMM-ADNN to solve the poplar image super-resolution problem. We have tested our algorithm on the MNIST dataset [14] and the 11K images of the Caltech-UCSD Birds-200-2011 (CUB-200-2011) dataset [28]. In the first two rows of Fig. 3, high resolution images, as shown in the last column, have been blurred (convolved) using a Gaussian kernel of size $3 \times 3$ and downsampled every 4 pixels in both vertical and horizontal directions to generate the corresponding low resolution images as shown in the first column. The bicubic interpolation of LR images and results using proposed Inf-ADMM-ADNN on a 20% held-out test set are displayed in column 2 and 3. Visually, the proposed Inf-ADMM-ADNN gives much better results than the bicubic interpolation, recovering more details including colors and edges. A similar task to super-resolution is motion deblurring, in which the convolution kernel is a directional kernel and there is no downsampling. The motion deblurring results using Inf-ADMM-ADNN are displayed in the bottom of Fig. 3 and are compared with the Wiener filtered deblurring result (the performance of Wiener filter has been tuned to the best by adjusting the regularization parameter). Obviously, the Inf-ADMM-ADNN gives visually much better results than the Wiener filter. Due to space limitations, more simulation results are available in supplementary materials (see Section 3.1 and 3.2).

To explore the convergence speed w.r.t. the ADMM regularization parameter $\beta$, we have plotted the normalized mean square error (NMSE) defined as NMSE $= \|\hat{\mathbf{x}} - \mathbf{x}\|_2^2 / \|\mathbf{x}\|_2^2$, of super-resolved MNIST images w.r.t. ADMM iterations using different values of $\beta$ in the right of Fig. 2. It is interesting to note that when $\beta$ is large, e.g., 0.1 or 0.01, the NMSE of ADMM updates converges to a stable value rapidly in a few iterations (less than 10). Reducing the value of $\beta$ slows down the decay of NMSE over iterations but reaches a lower stable value. When the value of $\beta$ is small enough, e.g., $\beta = 0.0001, 0.0005, 0.001$, the NMSE converges to the identical value. This fits well with the claim in Boyd's book [2] that when $\beta$ is too large it does not put enough emphasis on minimizing the

objective function, causing coarser estimation; thus a relatively small $\beta$ is encouraged in practice. Note that the selection of this regularization parameter is still an open problem.

### 4.3 Joint super-resolution and colorization

While image super-resolution tries to enhance spatial resolution from spatially degraded images, a related application in the spectral domain exists, i.e., enhancing spectral resolution from a spectrally degraded image. One interesting example is the so-called automatic colorization, i.e., hallucinating a plausible color version of a colorless photograph. To the best knowledge of the authors, this is the first time we can enhance both spectral and spatial resolutions from one single band image. In this section, we have tested the ability to perform joint super-resolution and colorization from one single colorless LR image on the celebA-dataset [16]. The LR colorless image, its bicubic interpolation and $\times 2$ HR image are displayed in the top row of Fig. 4. The ADMM updates in the 1st, 4th and 7th iterations (on held-out test set) are displayed in the bottom row, showing that the updated image evolves towards higher quality. More results are in the supplementary materials (see Section 3.3).

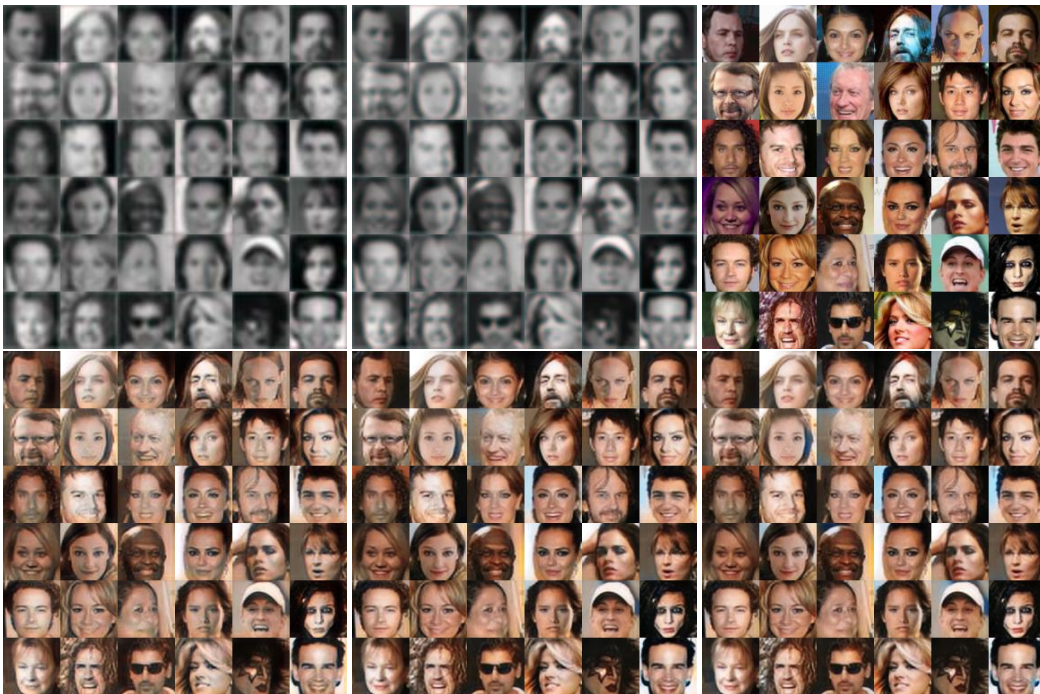

Figure 4: (top left) colorless LR image, (top middle) bicubic interpolation, (top right) HR ground-truth, (bottom left to right) updated image in **1**th, **4**th and **7**th ADMM iteration. Note that the colorless LR images and bicubic interpolations are visually similar but different in details noticed by zooming out.

## 5 Conclusion

In this paper we have proposed an accelerated alternating direction method of multipliers, namely, Inf-ADMM-ADNN to solve inverse problems by using two pre-trained deep neural networks. Each ADMM update consists of feed-forward propagation through these two networks, with a complexity of linear order with respect to the data dimensionality. More specifically, a conditional pixel shuffling denoising auto-encoder has been learned to perform amortized inference for the proximity operator. This auto-encoder leads to an implicit prior learned from training data. A data-independent structured convolutional neural network has been learned from noise to explicitly invert the big matrix associated with the forward model, getting rid of any inner loop in an ADMM update, in contrast to the conventional gradient based method. This network can also be combined with existing proximity operators to accelerate existing ADMM solvers. Experiments and analysis on both synthetic and real dataset demonstrate the efficiency and accuracy of the proposed method. In future work we hope to extend the proposed method to inverse problems related to nonlinear forward models.

# Appendices

We will address the question proposed by reviewers in this Appendix.

**To Reviewer 1**    The title has been changed to "An inner-loop free solution to inverse problems using deep neural networks" according to the reviewer's suggestion, which is in consistence with our arxiv submission. The pixel shuffling used in our PSDAE architecture is mainly to keep the filter size of every layer including input and output as the same, thus trick has been practically proved to remove the check-board effect. Especially for the super-resolution task with different scales of input/output, it is basically to use the input to regress the same scale output but with more channels.

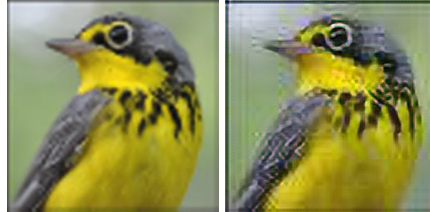

Figure 5: Result of super-resolution from SRGAN with different settings.

**To Reviewer 2**    As we explained in the rebuttal, we have the implementation of SRCNN with or without adversarial loss in our own but we did not successfully reproduce a reasonable result in our dataset. Thus, we did not include the visualization in the initial submission, since either blurriness or check-board effect will appear, but we will further fine-tune the model or use other tricks such as pixel shuffling. [11] has been added to the reference.

**To Reviewer 3**    Most of the questions have been addressed in the rebuttal.

## Acknowledgments

The authors would like to thank Siemens Corporate Research for supporting this work and thank NVIDIA for the GPU donations.

## Footnotes

[2]Note that this matrix inversion can be explicitly computed due to its small size in this toy experiment. In practice, this matrix is not built explicitly.

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
