[Supplementary Material]

# Supplementary Material for NIPS 2017 Paper # 1394: An Inner-loop Free Solution to Inverse Problems using Deep Neural Networks

**Kai Fai**[*]
Duke University
`kai.fan@stat.duke.edu`

**Qi Wei**[*]
Duke University
`qi.wei@duke.edu`

**Lawrence Carin**
Duke University
`lcarin@duke.edu`

**Katherine Heller**
Duke University
`kheller@stat.duke.edu`

## 1 Learning $A$ from training data

The objective to estimate $A$ is formulated as

$$\arg\min_A \sum_{i=1}^N \|\mathbf{y}_i - A\mathbf{x}_i\|_2^2 + \lambda\phi(A) \tag{1}$$

where $(\mathbf{x}_i, \mathbf{y}_i)_{i=1:N}$ are the training pairs and $\phi(A)$ corresponds to a regularization to $A$. Empirically, when $m$ is large enough, the regularization plays a less important role. The learned and real kernels for $A$ (of size $4 \times 4$) are visually very similar as is shown in Fig. 1.

Figure 1: (left) Ground-truth kernel for $A$ , (middle) learned kernel for $A$, (right) difference of these two.

---

[*]The authors contributed equally to this work.

Figure 2: (top) convolution matrix $H$, (middle) downsample matrix $S$, (right) strided convolution matrix $A = SH$.

## 2 Structure of matrix $A$ in Section 4.1

The degradation matrix $A$ in strided convolution can be decomposed as the product of $H$ and $S$, i.e., $A = SH$, where $H$ is a square matrix corresponding to 2-D convolution and $S$ represents the regular 2-D downsampling. In general, the blurring matrix $H$ is a block Toeplitz matrix with Toeplitz blocks. If the convolution is implemented with periodic boundary conditions, i.e., the pixels out of an image is padded with periodic extension of itself, the matrix $H$ is a block circulant matrix with circulant blocks (BCCB). Note that for 1-D case, the matrix $B$ reduces to a circulant matrix. For illustration purposes, an example of matrix $B$ for a 1-D case is given as below.

$$H = \begin{bmatrix} 0.5 & 0.3 & 0 & 0 & 0 & 0 & 0 & 0 & 0 & 0 & 0 & 0 & 0 & 0 & 0 & 0.2 \\ 0.2 & 0.5 & 0.3 & 0 & 0 & 0 & 0 & 0 & 0 & 0 & 0 & 0 & 0 & 0 & 0 & 0 \\ 0 & 0.2 & 0.5 & 0.3 & 0 & 0 & 0 & 0 & 0 & 0 & 0 & 0 & 0 & 0 & 0 & 0 \\ 0 & 0 & 0.2 & 0.5 & 0.3 & 0 & 0 & 0 & 0 & 0 & 0 & 0 & 0 & 0 & 0 & 0 \\ 0 & 0 & 0 & 0.2 & 0.5 & 0.3 & 0 & 0 & 0 & 0 & 0 & 0 & 0 & 0 & 0 & 0 \\ 0 & 0 & 0 & 0 & 0.2 & 0.5 & 0.3 & 0 & 0 & 0 & 0 & 0 & 0 & 0 & 0 & 0 \\ 0 & 0 & 0 & 0 & 0 & 0.2 & 0.5 & 0.3 & 0 & 0 & 0 & 0 & 0 & 0 & 0 & 0 \\ 0 & 0 & 0 & 0 & 0 & 0 & 0.2 & 0.5 & 0.3 & 0 & 0 & 0 & 0 & 0 & 0 & 0 \\ 0 & 0 & 0 & 0 & 0 & 0 & 0 & 0.2 & 0.5 & 0.3 & 0 & 0 & 0 & 0 & 0 & 0 \\ 0 & 0 & 0 & 0 & 0 & 0 & 0 & 0 & 0.2 & 0.5 & 0.3 & 0 & 0 & 0 & 0 & 0 \\ 0 & 0 & 0 & 0 & 0 & 0 & 0 & 0 & 0 & 0.2 & 0.5 & 0.3 & 0 & 0 & 0 & 0 \\ 0 & 0 & 0 & 0 & 0 & 0 & 0 & 0 & 0 & 0 & 0.2 & 0.5 & 0.3 & 0 & 0 & 0 \\ 0 & 0 & 0 & 0 & 0 & 0 & 0 & 0 & 0 & 0 & 0 & 0.2 & 0.5 & 0.3 & 0 & 0 \\ 0 & 0 & 0 & 0 & 0 & 0 & 0 & 0 & 0 & 0 & 0 & 0 & 0.2 & 0.5 & 0.3 & 0 \\ 0 & 0 & 0 & 0 & 0 & 0 & 0 & 0 & 0 & 0 & 0 & 0 & 0 & 0.2 & 0.5 & 0.3 \\ 0.3 & 0 & 0 & 0 & 0 & 0 & 0 & 0 & 0 & 0 & 0 & 0 & 0 & 0 & 0.2 & 0.5 \end{bmatrix}$$

An example of matrix $B$ for 2-D convolution of a $9 \times 9$ kernel with a $16 \times 16$ image is given in the top of Fig. 2. Clearly, in this huge matrix, a circulant structure is present in the block scale as well as within each block, which clearly demonstrates the self-similar pattern of BCCB matrix.

The downsampling matrix $S$ corresponds to downsampling the original signal and its transpose $S^T$ interpolates the decimated signal with zeros. Similarly, a 1-D example of downsampling matrix is shown in (2) for an illustrative purpose. An example of matrix $S$ for downsampling a $16 \times 16$ image to the size of $4 \times 4$, i.e., $S \in \mathbb{R}^{16 \times 256}$, is displayed in the middle of Fig. 2. The resulting degradation matrix $A$, which is the product of $S$ and $H$ is shown in the bottom of Fig. 2.

$$S = \begin{bmatrix} 1 & 0 & 0 & 0 & | & 0 & 0 & 0 & 0 & | & 0 & 0 & 0 & 0 & | & 0 & 0 & 0 & 0 \\ 0 & 0 & 0 & 0 & | & 1 & 0 & 0 & 0 & | & 0 & 0 & 0 & 0 & | & 0 & 0 & 0 & 0 \\ 0 & 0 & 0 & 0 & | & 0 & 0 & 0 & 0 & | & 1 & 0 & 0 & 0 & | & 0 & 0 & 0 & 0 \\ 0 & 0 & 0 & 0 & | & 0 & 0 & 0 & 0 & | & 0 & 0 & 0 & 0 & | & 1 & 0 & 0 & 0 \end{bmatrix} \tag{2}$$

## 3 More experimental results

### 3.1 Motion deblurring

The motion blurring kernel is shown in Fig. 3. More results of motion deblurring on held-out testing data for CUB dataset are displayed in Fig. 4, 5.

### 3.2 Super-resolution

More results of super-resolution on held-out testing data for CUB dataset are displayed in Fig. 6, 7.

### 3.3 Joint super-resolution and colorization

More results of joint super-resolution and colorization on held-out testing data for CelebA dataset are displayed in Fig. 8.

Figure 3: $9 \times 9$ motion blurring kernel.

(a) ground-truth image for motion deblurring task

(b) motion blurred images

Figure 4: motion blurring

(a) results using Wiener filter with the best performance by tuning regularization parameter.

(b) deblurred results using Inf-ADMM-ADNN

Figure 5: motion deblurring

(a) LR images

(b) bicubic interpolations

Figure 6: super-resolution

(a) super-resolved ($\times 4$) images using Inf-ADMM-ADNN

(b) HR groundtruth

Figure 7: super-resolution

(a) colorless images

(b) joint super-resolution ($\times 2$) and colorization using Inf-ADMM-ADNN

(c) HR groundtruth

Figure 8: joint super-resolution and colorization

# 4 Networks Setting

## 4.1 Network for updating x

For MNIST dataset, we did not use the pixel shuffling strategy, since each data point is a $28 \times 28$ grayscale image, which is relatively small. Alternatively, we used a standard denoising auto-encoder with architecture specifications in Table 1.

Table 1: Network Hyper-Parameters of DAE for MNIST

| Input Dim | Layer | Output Dim |
|---|---|---|
| $28 \times 28 \times 1$ | Conv$(5, 5, 1, 32)$-Stride$(2, 2)$-'SAME'-Relu | $14 \times 14 \times 32$ |
| $14 \times 14 \times 32$ | Conv$(5, 5, 32, 64)$-Stride$(2, 2)$-'SAME'-Relu | $7 \times 7 \times 64$ |
| $7 \times 7 \times 64$ | Conv$(5, 5, 64, 128)$-Stride$(2, 2)$-'VALID'-Relu | $2 \times 2 \times 128$ |
| $2 \times 2 \times 128$ | Conv$(2, 2, 128, 64)$-Stride$(1, 1)$-'VALID'-None | $1 \times 1 \times 64$ |
| $1 \times 1 \times 64$ | Conv_trans$(3, 3, 64, 128)$-Stride$(1, 1)$-'VALID'-Relu | $3 \times 3 \times 128$ |
| $3 \times 3 \times 128$ | Conv_trans$(5, 5, 128, 64)$-Stride$(1, 1)$-'VALID'-Relu | $7 \times 7 \times 64$ |
| $7 \times 7 \times 64$ | Conv_trans$(5, 5, 64, 32)$-Stride$(2, 2)$-'SAME'-Relu | $14 \times 14 \times 32$ |
| $14 \times 14 \times 32$ | Conv_trans$(5, 5, 32, 1)$-Stride$(2, 2)$-'SAME'-Sigmoid | $28 \times 28 \times 1$ |

For CUB-200-2011 dataset, we applied a periodical pixel shuffling layer to the input image of size $256 \times 256 \times 3$ with the output of size $64 \times 64 \times 48$. Note that we did not use any stride here since we keep the image scale in each layer identical. The architecture of the cPSDAE is given in Table 2. For CelebA dataset, we applied the periodical pixel shuffling layer to the input image of size $128 \times 128 \times 3$ with the output of size $32 \times 32 \times 48$, and the rest of setting is the same as CUB-200-2011 dataset, as shown in Table 3. In terms of the discriminator, we fed the pixel shuffled images. The architecture of the disriminator is the same as the one in DCGAN.

Table 2: Network hyper-parameters of cPSDAE for CUB-200-2011

| Input Dim | Layer | Output Dim |
|---|---|---|
| $256 \times 256 \times 3$ | periodical pixel shuffling | $64 \times 64 \times 48$ |
| $64 \times 64 \times 48$ | Conv$(4, 4, 48, 128)$-'SAME'-Batch_Norm-Relu | $64 \times 64 \times 128$ |
| $64 \times 64 \times 128$ | Conv$(4, 4, 128, 64)$-'SAME'-Batch_Norm-Relu | $64 \times 64 \times 64$ |
| $64 \times 64 \times 64$ | Conv$(4, 4, 64, 32)$-'SAME'-Batch_Norm-Relu | $64 \times 64 \times 32$ |
| $64 \times 64 \times \{32, 3\}$ | Concatenate in Channel | $64 \times 64 \times 35$ |
| $64 \times 64 \times 35$ | Conv$(4, 4, 35, 64)$-'SAME'-Batch_Norm-Relu | $64 \times 64 \times 64$ |
| $64 \times 64 \times 64$ | Conv$(4, 4, 64, 128)$-'SAME'-Batch_Norm-Relu | $64 \times 64 \times 128$ |
| $64 \times 64 \times 128$ | Conv$(4, 4, 128, 48)$-'SAME'-Batch_Norm-Relu | $64 \times 64 \times 48$ |
| $64 \times 64 \times 48$ | periodical pixel shuffling | $256 \times 256 \times 3$ |

Table 3: Network hyper-parameters of cPSDAE for CelebA

| Input Dim | Layer | Output Dim |
|---|---|---|
| $64 \times 64 \times 3$ | periodical pixel shuffling | $32 \times 32 \times 12$ |
| $32 \times 32 \times 12$ | Conv$(4, 4, 12, 128)$-'SAME'-Batch_Norm-Relu | $32 \times 32 \times 128$ |
| $32 \times 32 \times 128$ | Conv$(4, 4, 128, 64)$-'SAME'-Batch_Norm-Relu | $32 \times 32 \times 64$ |
| $32 \times 32 \times 64$ | Conv$(4, 4, 64, 32)$-'SAME'-Batch_Norm-Relu | $32 \times 32 \times 32$ |
| $32 \times 32 \times \{32, 3\}$ | Concatenate in Channel | $32 \times 32 \times 35$ |
| $32 \times 32 \times 35$ | Conv$(4, 4, 35, 64)$-'SAME'-Batch_Norm-Relu | $32 \times 32 \times 64$ |
| $32 \times 32 \times 64$ | Conv$(4, 4, 64, 128)$-'SAME'-Batch_Norm-Relu | $32 \times 32 \times 128$ |
| $32 \times 32 \times 128$ | Conv$(4, 4, 128, 12)$-'SAME'-Batch_Norm-Relu | $32 \times 32 \times 12$ |
| $32 \times 32 \times 12$ | periodical pixel shuffling | $64 \times 64 \times 3$ |

## 4.2 Network for updating z

As described in Section 3.2, the neural network to update $\mathbf{z}$ was designed to have symmetric architecture. The details of this architecture is given in Table 4. Note that $W \times H$ represents the size of the width and height of measurement $\mathbf{y}$.

Table 4: Symmetric network hyper-parameters for updating $\mathbf{z}$

| Input Dim | Layer | Output Dim |
|---|---|---|
| $H \times W \times 3$ | Conv_trans(4,4,3,32, $W_0$)-'SAME'-Relu | $H \times W \times 32$ |
| $H \times W \times 32$ | Conv_trans(4,4,32,64, $W_1$)-'SAME'-Relu | $H \times W \times 64$ |
| $H \times W \times 64$ | Conv(4,4,3,32, $W_1$)-'SAME'-Relu | $H \times W \times 32$ |
| $H \times W \times 32$ | Conv(4,4,32,64, $W_0$)-'SAME' | $H \times W \times 3$ |