[Reviews · NeurIPS 2017]

Reviewer 1



An interesting paper that solves linear inverse problems using a combination of two networks: one that learns a proximal operator to the signal class of interest, and the other that serves as a proxy for a large scale matrix inversion. The proximal operator is reusable whenever the signal domain is unchanged. One would need to retrain only the matrix inversion network when the underlying problem is changed. This is a significant advantage towards reusability of the training procedure. Strengths + Novelty + A very interesting problem Weaknesses - An important reference is missing - Other less important references are missing - Bare-bones evaluation The paper provides an approach to solve linear inverse problems by reducing training requirements. While there is some prior work in this area (notably the reference below and reference [4] of the paper), the paper has some very interesting improvements over them. In particular, the paper combines the best parts of [4] (decoupling of signal prior from the specific inverse problem being solved) and Lista (fast implementation). That said, evaluation is rather skim — almost anecdotal at times — and this needs fixing. There are other concerns as well on the specific choices made for the matrix inversion that needs clarification and justifications. 1) One of the main parts of the paper is a network learnt to invert (I + A^T A). The paper used the Woodbury identity to change it to a different form and learns the inverse of (I+AA^T) since this is a smaller matrix to invert. At test time, we need to apply not just this network but also "A" and "A^T" operators. A competitor to this is to learn a deep network that inverts (I+A^T A). A key advantage of this is that we do not need apply A and A^T during test time. It is true that that this network learns to inverts a larger matrix ... But at test time we have increased rewards. Could we see a comparison against this ? both interms of accuracy as well as runtime (at training and testing) ? 2) Important reference missing. The paper is closely related to the idea of unrolling, first proposed in, “Lista” http://yann.lecun.com/exdb/publis/pdf/gregor-icml-10.pdf While there are important similarities and differences between the proposed work and Lista, it is important that the paper talks about them and places itself in appropriate context. 3) Evaluation is rather limited to a visual comparison to very basic competitors (bicubic and wiener filter). It would be good to have comparisons to - Specialized DNN: this would provide the loss in performance due to avoiding the specialized network. - Speed-ups over [4] given the similarity (not having this is ok given [4] is an arXiv paper) - Quantitative numbers that capture actual improvements over vanilla priors like TV and wavelets and gap to specialized DNNs. Typos - Figure 2: Syntehtic

Reviewer 2



This paper proposes an efficient method for applying ADMM to solve image processing problems involving neural networks. The proposed method avoids solving large matrix inversions directly. I have a few criticisms of this work: 1) The authors should discuss the relationship between their proximal operators, and those used in 4. A main contribution of this paper is the use of a neural network to replace the proximal operator, but this was already done in [4], although with different methods. 2) I'm unclear about the details of the synthetic experiments. It seems like you trained a neural network to replicate the behavior of the shrink operator? I think? Also, It's unclear how the runtime of the network implementation is comparable to the shrink operator. 3) Major weakness of this paper: they don't compare to any other modern methods (they compare to simple things like a Weiner filter, but this is a *really* low bar and easy to beat). What about other variational deblurring method (like TV or wavelet based methods), and what the methods discussed in [4], which can handle the super-resolution problems posed here. I like the ideas in this paper, but the experimental comparisons just seem fairly weak, which is why I can't strongly recommend this paper for publication. note: I mentioned reference [4] several times because of its close relationship to the work under review. I am not an author on that paper, nor do I have any direct affiliations with the authors or their institutions.

Reviewer 3



In recent literature, there are studies showing that the testing-time inference can be approximated by the forward-pass of deep neural network. This paper falls into this category. In particular, two neural networks are proposed to approximate the inner loops of the ADMM arising in the inverse image processing problems (e.g., super resolution and deblurring). Techniques and devises are sound; empirical results are satisfactory. There is minor concern about the title, which seems over-claiming/mis-leading: the proposed approach is data-dependent and not really aims at general ADMM numeric optimization. Another minor suggestion is that more expositions can be used to justify the cPSDAE (discussions at line 126 and line 206; at least make the contents self-contained here).